# Postoperative Impact of Pontocerebellar Angle Surgery on the Quality of Life in Patients with Vestibular Schwannoma

**Valentina Foscolo** [1], **Luigi de Gennaro** [2], **Alessandra Murri** [1], **Luca Speranzon** [2], **Francesco Signorelli** [2], **Nicola Quaranta** [1,*] **and Raffaella Messina** [2]

1   Otolaryngology Unit, Department of Translational Biomedicine and Neurosciences (DiBraiN), University "Aldo Moro" of Bari, 70124 Bari, Italy
2   Division of Neurosurgery, Department of Translational Biomedicine and Neurosciences (DiBraiN), University "Aldo Moro" of Bari, 70124 Bari, Italy
*   Correspondence: nicolaantonioadolfo.quaranta@uniba.it

**Abstract:** Background: Vestibular Schwannomas are benign tumors arising from the VIII CN. Surgical treatment is indicated in case of tumors larger than 2.5 cm in the cerebellopontine angle or in the case of cranial nerve dysfunction. The aim of the present study was to evaluate the QoL by means of the PANQOL questionnaire in a group of surgically treated patients mainly affected by large and giant VS Methods: All patients underwent preoperative and postoperative otoneurological evaluation and gadolinium enhanced MRI and they completed, independently, the PANQOL questionnaire at last follow up. Results: 70% of patients presented with large Koos III or IV VS Each domain of PANQOL showed a strong correlation with the total PANQOL score. In relation to the postoperative facial nerve function, patients with poorer function showed significantly lower score in the facial dysfunction and pain, patients with postoperative balance problems showed a significantly lower PANQOL score for domains of balance and pain. Conclusions: This study showed that postoperative QoL of patients was acceptable even if there were some domains that were more affected, such as hearing and balance domains; therefore, the lowest scores suggest the need for vestibular rehabilitation programs and strategies that improve postoperative hearing.

**Keywords:** vestibular schwannoma; acoustic neuroma; quality of life; questionnaires; Penn Acoustic Neuroma Quality of Life (PANQOL); microsurgery; translabyrinthine; hearing and balance; post operative facial function

## 1. Introduction

Vestibular schwannomas (VS) are benign, slow-growing tumors arising from the VIII CN and constitute 8% of all intracranial neoplasms and 90% of cerebellopontine angle (CPA) lesions [1].

Due to the slow growth of these tumors, treatment options include surgical excision, stereotactic radiosurgery (SRS), and conservative management/active surveillance. Individual patient management depends on various factors including age, medical comorbidities, size and location of the tumor, and hearing status [2].

Surgical treatment is generally indicated in the case of tumors larger than 2.5 cm in the cerebello-pontine angle with the primary aims of complete tumor removal and the preservation of facial function and the patient's quality of life (QoL) [3].

In Italy, the preferred surgical approaches for VS removal are the translabyrinthine (TL) and retrosigmoid (RS) [3]. TL is a presigmoid transmastoid approach that allows the exposure of the internal auditory canal (IAC) and cerebellopontine angle (CPA) after the removal of the posterior labyrinth and the presigmoid bone. It is therefore the best option when hearing preservation is not an issue and, compared to the RS approach, allows the identification of the facial nerve both at the root entry zone of the brainstem and at the

fundus of the IAC. The RS approach is mainly an intradural approach that allows a large view of the CPA; however, cerebellar retraction is needed, and the fundus of the IAC is difficult to expose especially in the case of hearing preservation with the risks of subtotal removal [4].

The evaluation of QoL in VS patients has become increasingly important in recent years. In a systematic review, Barker-Collo et al. [2] have reported that surgical treatment does not improve the QoL in patients affected by small-to-medium size tumors; therefore, initial observation has been proposed as the first therapeutic option in these patients [1]. In the case of large and giant VS, surgical treatment represents the only therapeutic option, especially in the case of cranial nerve dysfunction [3] and therefore the evaluation of the surgical results as well as the post-operative QoL must be considered at the time of surgical planning. The Penn Acoustic Neuroma Quality of Life Questionnaire (PANQOL) is a disease-specific tool proposed by Shaffer et al. [5] that measures the QoL of VS patients, evaluating the effect of the tumor and of the treatment in six specific domains: balance, energy, hearing, anxiety, face, general health and pain. Lucidi et al. [6] have recently adapted the questionnaire in Italian and have evaluated the QoL of VS patients treated with three surgical techniques. The aim of the present study was to evaluate the QoL by means of the PANQOL questionnaire in a group of surgically treated patients mainly affected by large and giant VS; in addition, the internal consistency and reliability of the Italian PANQOL questionnaire and factors that may predict patients' QoL were evaluated.

## 2. Materials and Methods

### 2.1. Participants

Between April 2018 and January 2022, 31 patients affected by VS underwent microsurgical tumor removal and represent the study group. Patients affected by skull base pathologies other than VS, patients with neurofibromatosis type 2, those who had received multiple active treatments, or those who had undergone previous microsurgical tumor removal were excluded.

A retrospective chart review was conducted for the included patients containing preoperative (sex, age, hearing impairment, tumor side and size), intra-operative (surgical approach, grade of resection, time of surgery) and post-operative characteristics and symptoms (facial paresis, balance problems, postoperative complications).

### 2.2. Procedures

All patients underwent pre-operative and post-operative evaluation consisting of clinical history, complete otoneurological evaluation of the cranial nerves, vestibular bed side examination (spontaneous nystagmus evaluation, Romberg test, Unterberger test, Head Shaking Test and clinical Head Impulse Test), tonal and speech audiometry and gadolinium enhanced MRI.

Tumor size was classified according to Koos et al. [7] in four stages. Facial function was classified according to the House–Brackmann scale (HB) [8], while pre-operative hearing was classified according to the classification system of the Committee on Hearing and Equilibrium of the American Academy of Otolaryngology-Head and Neck Surgery (AAO-HNS) 1995 [9]. Tumor removal was classified in terms of the percentage of tumor removed by resection as: gross total resection (no macroscopic residual of tumor resection 100%); near total resection (the residual consists of only a small, thin capsular peel, <25 mm$^2$, <2 mm thick); subtotal resection (a substantial portion of tumor remains >25 mm$^2$, >2 mm thick removal) as proposed by Bloch et al. [10]. Complications were classified as intra-operative or post-operative. Average follow-up was 8 months (range 6–18 months).

The Italian version of the PANQOL questionnaire was used [6]. PANQOL comprises 26 multiple-choice questions that focus on the following areas: balance (six items), energy (six items), hearing (four items), anxiety (four items), face (three items), general health (two items) and pain (one item). Patients were asked to rate each item from 1 (strongly disagree) to 5 (strongly agree). A total instrument score was calculated as the unweighted

average of the domain scores and reported on a scale from 0 to 100 (worst to best QoL). The questionnaires were completed by the patients independently in our clinics at last follow-up.

### 2.3. Statistical Analyses

Descriptive statistics of the PANQOL questionnaire were performed. Reliability was measured with one measure tool Cronbach's Alpha. A domain correlation matrix with Spearman coefficient was created and a subgroup analysis was performed with measures such as the Wilcoxon rank-sum test and Kruskal–Wallis test. Nonparametric tests were utilized considering the lack of the normal distribution of the domain scale. A Bonferroni correction was performed in the case of multiple comparisons. Data were considered statistically significant with a p value ($\alpha$) < 0.05. All statistical analyses were performed using the software SPSS statistics (IBM-1 New Orchard Road, Armonk, NY, USA) version 28.0.1.1.

## 3. Results

### 3.1. Patients Characteristics

Thirty-one patients undergoing surgery for VS between April 2018 and January 2022 were included in this study. Fifteen were female and 16 were male with a median age of 54.2 years (range, 16–79 year). Sixteen VS were located on the left side, while 15 were found on the right side. Seventy percent of patients presented with large Koos III or IV VS (Table 1).

**Table 1.** Demographics and clinical characteristics.

| Characteristics | Value | % |
|---|---|---|
| Number of patients | 31 | 100 |
| Gender | 31 | 100 |
| Male | 15 | 49 |
| Female | 16 | 51 |
| Age | | |
| Median | 54.2 | |
| Range | 16–79 | |
| Koos grade | | |
| I (intracanalar) | 1 | 3.2 |
| II (CPA not filled) | 8 | 25.8 |
| III (CPA filled) | 10 | 32.3 |
| IV (displacement) | 12 | 38.7 |

Twenty-six patients underwent TL approach and five patients the RS approach; resection was gross total in 23 patients and near total in eight (Table 2).

**Table 2.** Surgical notes.

| Characteristics | Value | % |
|---|---|---|
| Number of patients | 31 | 100 |
| Surgical approach | 31 | 100 |
| Translabyrinthine | 26 | 84 |
| Translabyrinthine (Koos I–II) | 9 | 35 |
| Translabyrinthine (Koos III–IV) | 17 | 65 |
| Retrosigmoid | 5 | 16 |
| Retrosigmoid (Koos I–II) | 0 | 0 |
| Retrosigmoid (Koos III–IV) | 5 | 100 |

**Table 2.** *Cont.*

| Characteristics | Value | % |
|---|---|---|
| Resection | | |
| Total | 23 | 74.2 |
| Translabyrinthine | 19/26 | 61.3 |
| Translabyrinthine (Koos I–II) | 9 | 29 |
| Translabyrinthine (Koos III–IV) | 10 | 32.2 |
| Retrosigmoid | 4/5 | 12.9 |
| Retrosigmoid (Koos III–IV) | 4 | 12.9 |
| Near total (<25 mm$^2$, <2 mm thick) | 8 | 25.8 |
| Translabyrinthine | 7 (7/26) | 22.6 |
| Translabyrinthine (Koos III–IV) | 7 | 22.6 |
| Retrosigmoid | 1 (1/5) | 3.2 |
| Retrosigmoid (Koos III–IV) | 1 | 3.2 |
| Surgical complication | | |
| Mortality | 0 | 0 |
| Hydrocephalus | 2 | 6.4 |
| CSF leak | 1 | 3.2 |
| Pulmonary embolism | 3 | 9.7 |
| Jugular vein and transverse sinus thrombosis | 1 | 3.2 |
| Cerebellar ischemia | 2 | 6.4 |

All patients presented with a normal pre-operative facial nerve function, while post-operative facial function evaluated at last follow-up is reported in Table 3. Of the patients, 61.3% presented a facial function HB grade 1 or 2; 22.6% grade 3 and 16.1% worse than grade 3.

**Table 3.** Post-operative facial function.

| House–Brackmann Scale | Value | % |
|---|---|---|
| Grade I | 9 | 29 |
| Grade II | 10 | 32.3 |
| Grade III | 7 | 22.6 |
| Grade IV | 3 | 9.7 |
| Grade V | 0 | 0 |
| Grade VI | 2 | 6.4 |

At last follow-up, all patients presented with vestibular areflexia in the operated side, and 12 (38.7%) reported balance problems. Otoneurological examinations showed cerebellar ataxia in four cases (all Koos III–IV), central nystagmus in two cases (all Koos III–IV) and postural instability in the remaining six cases (three cases Koos I–II and three cases Koos III–IV).

Mortality was 0% and no intra-operative complications occurred. Nine patients developed post-operative complications. In two cases, additional surgery (a lumbo-peritoneal and a ventriculo-peritoneal shunt) was needed because of post-operative hydrocephalus (both Koos IV). In one case, a CSF leak was managed with temporary lumbar drainage, while all other complications (pulmonary embolism and cerebellar ischemia) were successfully managed with medical treatment (Table 2).

*3.2. PANQOL Results*

Internal consistency was measured by Cronbach's alpha and was high for all domains (balance: 0.761; hearing: 0.778; anxiety: 0.74; facial dysfunction: 0.81; energy: 0.748; pain: 0.818; general health: 0.795). Domain scores were calculated on a scale between 0 and 100 as previously described (0 1/4 worst to 100 1/4 best QOL), and a total instrument score was calculated as the equal average of all domain scores (10):

$$x = (\text{actual raw value} - \text{lowest possible raw value})/\text{possible range of raw value} \times 100 \qquad (1)$$

Table 4 reports the average scores (with SD) and the range (min and max) for each domain and for the total score. Average scores were greater than 50 for balance, anxiety, facial dysfunction, energy and pain, while they were lower than 50 for hearing and general health.

**Table 4.** Descriptive statistics of PANQOL questionnaire (*n* = 31).

| Item | N | Minimum | Maximum | Average | SD |
|---|---|---|---|---|---|
| PANQOL total | 31 | 11.0 | 93.3 | 56.135 | 20.4501 |
| Balance | 31 | 0.00 | 100.00 | 50.9571 | 32.86743 |
| Hearing | 31 | 6.25 | 87.50 | 38.0065 | 20.24861 |
| Anxiety | 31 | 18.75 | 100.00 | 65.1371 | 26.58515 |
| Facial dysfunction | 31 | 8.34 | 100.00 | 67.6142 | 24.48965 |
| Energy | 31 | 12.50 | 100.00 | 62.8139 | 26.92814 |
| Pain | 31 | 0 | 100 | 66.94 | 37.298 |
| General health | 31 | 0.0 | 100.00 | 39.919 | 30.0034 |

A non-parametric correlation test (Spearman correlation matrix) on each domain of PANQOL questionnaire was performed. All domains showed a strong correlation with the total PANQOL score and several showed a robust correlation with others (Table 5).

**Table 5.** Rho Spearman correlation matrix for PANQOL domains score.

| | Balance | Hearing | Anxiety | Facial Dysfunction | Energy | Pain | General Health | Total |
|---|---|---|---|---|---|---|---|---|
| Balance | 1 | - | - | - | - | - | - | - |
| Hearing | 0.5 * | 1 | - | - | - | - | - | - |
| Anxiety | 0.53 * | 0.41 * | 1 | - | - | - | - | - |
| Facial Dysfunction | 0.41 | 0.23 | 0.43 | 1 | - | - | - | - |
| Energy | 0.46 | 0.55 * | 0.64 * | 0.32 | 1 | - | - | - |
| Pain | 0.23 | 0.41 | 0.54 * | 0.14 | 0.67 * | 1 | - | - |
| General Health | 0.45 | 0.27 | 0.52 * | 0.21 | 0.59 * | 0.21 | 1 | - |
| Total | 0.79 * | 0.72 * | 0.77 * | 0.48 * | 0.8 * | 0.48 * | 0.6 * | 1 |

* $p < 0.05$.

A single sample analysis was performed in order to investigate factors influencing post-operative PANQOL score. The following factors were evaluated: pre-operative tumor size, pre-operative hearing level, post-operative facial dysfunction, post-operative balance, and post-operative complications. The type of surgical approach was not evaluated because of the small number of patients undergoing RS removal.

In relation to the pre-operative size, median PANQOL score was lower in patients with Koos III and IV VS (52.9) compared to Koos I and II (59); however, the difference was not significant. The "general health" and "facial dysfunction" domain scores were also not significantly lower in Koos III and IV. In relation to the post-operative facial nerve function, patients with poorer function showed significantly lower scores in the "facial dysfunction" ($p = 0.032$) and "pain" ($p = 0.014$) domains, while the total scores and all other domains were not significantly different. Twelve patients with post-operative balance problems showed a significantly lower PANQOL score for the domains of "balance" ($p = 0.019$) and "pain" ($p = 0.033$), while statistical significance was reached for the "facial dysfunction" domain ($p = 0.049$) without Bonferroni correction for multiple comparisons (Table 6). Pre-operative hearing level as well as the presence of post-operative complications did not correlate with post-operative QoL.

**Table 6.** PANQOL scores according to the presence of post-operative balance disturbance.

| Item | Balance Problems | | No Balance Problems | | KW |
|---|---|---|---|---|---|
| | Median | SD | Median | SD | $p$ ($\alpha$) |
| Anxiety | 62.5 | 24.37 | 62.5 | 30.87 | n.s. |
| Facial Dysfunction | 55.15 | 61.97 | 66.7 | 29.33 | 0.049 ** |
| General health | 37.5 | 33.69 | 37.5 | 22.82 | n.s. |
| Balance | 33.33 | 31.69 | 50 | 35.86 | 0.019 * |
| Hearing | 37.25 | 21.73 | 43.75 | 17.97 | n.s. |
| Energy | 83.4 | 24.52 | 45.9 | 27.22 | n.s. |
| Pain | 50 | 41.00 | 75 | 27.87 | 0.033 * |
| Total | 52.9 | 18.05 | 55.5 | 24.54 | n.s. |

* Statistically significant after Bonferroni correction. ** Statistically significant without Bonferroni correction. n.s. not significant; SD, standard deviation. Wilcoxon test was used ($p < 0.05$).

## 4. Discussion

The evaluation of the QoL in patients undergoing different types of treatment has become increasingly important in the otological and otoneurological literature.

In recent years, disease-specific questionnaires have been proposed in order to improve the measurement of specific diseases or their treatments on the patient's QoL. The process of translation and validation of these questionnaires in different languages allowed the comparison of results and the evaluation of the impact of specific symptoms on QoL in different populations.

For example, the validation of the Chronic Otitis Media Questionnaire-12 (COMQ-12) in different languages including Italian has allowed the evaluation of the effect of chronic otitis media on patients' QoL in different populations [11].

The PANQOL is a disease-specific tool that has been increasingly used over the years to evaluate the QoL of VS patients and has been translated and validated in several languages including Italian [6]. In the present study, the Italian version of the PANQOL questionnaire presented high Cronbach's alpha, as has also been reported by other authors in different languages [5,12–15], confirming its good reliability as a specific tool for analyzing the quality of life of VS patients.

In the present series, VS patients presented with Koos III and IV tumors in 71% of cases, representing a typical population of a multidisciplinary otolaryngologic and neurosurgical clinic. In fact, while smaller tumors have different therapeutic options, larger VS need surgical treatment and therefore the evaluation of the post-operative QoL has become increasingly important in patient counselling.

In comparison with a "normal population," as well as with VS patients that are conservatively managed, the QoL of surgically treated patients has been reported to be poorer [1,2,16,17]. Selection bias has been reported since patients that undergo a watch-and-wait protocol usually present with smaller tumors and better hearing compared to those undergoing surgery. In cases of larger VS, even with the use of a not-disease-specific questionnaire such as the SF-36, Turel et al. [18] have reported that patients affected by large and giant VS may improve their QoL after surgery. The results of the present study showed that post-operative QoL of patients was acceptable even if there are some domains that were more affected such as the general health, hearing and balance domains.

While hearing and balance also represent the most affected domains in other series of surgically treated patients [12,13,16], general health was reported to be significantly impaired only by Pruijn et al. [16]. Such non-homogeneous results have been attributed to the fact that only two questions explore the general health domain in the PANQOL questionnaire and lead to a poor internal consistency of this scale [12]. The removal of the VS together with the drilling of the labyrinth in the case of the TL approach induce complete unilateral vestibular ablation and hearing loss. A single side deafness has been associated with poor sound localization and speech discrimination especially in noisy environments [19]; in addition, the hearing level of the contralateral ear represents a crucial factor in speech discrimination [20]. In the present series, mean age at surgery was

54.1 years, suggesting that more than half of the patients may possibly present with some degree of age-related hearing loss [21,22]. Even if hearing can be rehabilitated with CROS systems and bone anchored hearing aids, these aids restore a pseudo-binaural hearing that does not improve speech discrimination [23] nor sound-localization [24]. Cochlear implantation is the only device that is able to restore binaural hearing, but is not feasible in most VS surgeries due to the cochlear nerve damage induced by the tumor [25]. The poor results obtained in the balance domain are instead associated with the complete ablation of the vestibular function after TL approach and VS removal; even if a central compensation is expected after surgery, not all patients obtain a good balance and several prognostic factors have been reported [26], among them the size of the tumor and vestibular rehabilitation. In the present series, most of the patients presented with large VS and none underwent vestibular rehabilitation, suggesting the need for post-operative rehabilitation programs.

In patients operated on for small tumors, delayed compensation occurred, while in patients operated on for larger tumors, both delayed compensation and central vertigo occurred. As reported by other authors [27], early vestibular rehabilitation should be implemented in all patients and particularly in the elderly and those affected by large VS

The facial dysfunction domain was associated with the least impact on QoL. In the present series as reported by other authors [1,12,16], there was a high correlation between post-operative facial nerve function according to the HB classification and the self-reported facial dysfunction PANQOL domain. Although the majority of the patients presented with large tumors, 61.3% presented with grade 1 or 2 facial function and only 6% of patients presented with grade 6. Near total removal was performed in eight cases in order to preserve the facial nerve anatomy and function and has been proposed as the standard of care when the facial nerve anatomy is at risk, especially in the case of large VS [28].

Anxiety, energy and pain domains, together with facial dysfunction, presented the highest scores and therefore in the present series do not represent factors that impair the QoL. The low incidence of reported headache was associated with the number of TL approach that, compared to RS, has been shown to be less associated with post-operative headache [29].

In the present series, no pre-operative factor was associated with post-operative QoL decrease; in particular, size did not influence the post-operative QoL in any domain. Similar results were reported by other authors [12,30,31], supporting a conservative management in smaller tumors and suggesting active treatment for larger tumors [13]. Finally, as also reported by Glaas et al. [12], patients with post-operative balance problems had lower scores in the balance domain as well as pain and facial dysfunction, corroborating the need for post-operative vestibular rehabilitation programs.

Although the present study presents many limitations, such as the limited sample and the absence of a control group, it evaluates the QoL of a group of Italian patients affected by large VS and surgically treated mainly by the TL approach. The low scores obtained in the hearing and balance domains suggest the need for specific rehabilitation programs and for strategies that improve post-operative hearing. Specific tools aimed at the evaluation of the hearing dysfunction as well as therapeutic strategies aimed at the restoration of the binaural hearing should be implemented.

## 5. Conclusions

The evaluation of the QoL in VS patients is important in the otological and neurotological literature. This study showed that the post-operative QoL of patients was acceptable even if there were some domains that were more affected, such as the hearing and balance domains. The lowest scores suggest the need for rehabilitation programs and strategies that improve post-operative hearing.

**Supplementary Materials:** The following supporting information can be downloaded at: https://www.mdpi.com/article/10.3390/audiolres12060061/s1.

**Author Contributions:** Conceptualization, N.Q. and F.S.; methodology, V.F. and L.d.G.; statistical methods, L.d.G.; validation, N.Q., F.S. and R.M.; formal analysis, V.F. and L.d.G.; investigation, V.F., L.d.G., A.M. and L.S.; resources, V.F. and L.d.G.; data curation, V.F. and L.d.G.; writing—original draft preparation, V.F. and L.d.G.; writing—review and editing, V.F., L.d.G., R.M. and N.Q.; supervision, R.M., N.Q. and F.S. All authors have read and agreed to the published version of the manuscript.

**Funding:** This research received no external funding.

**Institutional Review Board Statement:** This study was approved by the University of Bari institutional research board (Approval code #7191) and it was conducted in accordance with the Principles of Ethics for Medical Research Involving Human Subjects set in the Declaration of Helsinki and its subsequent amendments.

**Informed Consent Statement:** Informed consent was obtained from all subjects involved in the study.

**Data Availability Statement:** The statistical analysis plan, study protocol, and informed consent of the included patients are present in the Supplementary File; identified data and raw data are available upon motivated request to the corresponding author (nicolaantonioadolfo.quaranta@uniba.it) and may be reused to reproduce research, to make public assets available to the public, to leverage investments in research, and to advance research and innovation.

**Conflicts of Interest:** The authors declare no conflict of interest.

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
