# Peer review of "Postoperative Impact of Pontocerebellar Angle Surgery on the Quality of Life in Patients with Vestibular Schwannoma"

_audiolres, doi:10.3390/audiolres12060061_

Round 1
Reviewer 1 Report
General recommendations for small changes:
I suggest to incorporate surgical complications into the Table 2 Surgical notes. It would be more clear and easier to grasp. Also, the complications should be expressed as a percent, not only the absolute numbers.
The results of the bed side neurotological examination are not presented, even if they are crucial in light of the high correlation cited between OTN results and PANQOL Balance domain. Add another table?
Koos classification should be cited consistently – in the paper scoring is written in Roman numerals (I,II,III,IV – usual convention) or by Arabic numerals (1, 2, 3, 4) in other parts of the text and tables.
Also , in tables the notation „KOOS“ is used. Koos is an author s name, not an abbreviation!.
PANQOL results should be interpreted also in respect to MCID – Minimal Clinically Important Difference value as well . See eg. :
Carlson, M. L. et al. The Minimal Clinically Important Difference in Vestibular Schwannoma Quality-of-Life Assessment. Otolaryngology Head Neck Surg 153, 202–208 (2015).
On the following lines typing errors occur:
row 72
typing error – Participants, not Partecipants
row 85
typing error – Untemberger
row 89
House Brackmann, should be House-Brackmann scale
Table 3:
should be House-Brackmann
rows description (Grade I…) should be shifted one line down
rows 248-249
a word missing in the sentence:
In the present series no pre-operative factor was associated with post-operative
QoL ?decrease?, in particular size did not influence the post-operative QoL in any domain.
row 264
typing error: The evaluation of the QoL in VS patients is important in the ontological (should be otological) and neurotological literature.
row 144 : the reference (10) does not seem to be relevant in the context – error?
Reviewer 2 Report
1、The authors can compare the preoperative and postoperative conditions. Whether the postoperative situation will be different for patients with preoperative hearing loss or vestibular function impairment.
2、The author can observe the changes of patients' scores in different periods after operation.
3、Observe whether rehabilitation treatment is helpful to alleviate the discomfort of patients. Observe whether cognitive education can change the scoring results.
